# Assembly of a Benthic Microbial Community in a Eutrophic Bay with a Long History of Oyster Culturing

**DOI:** 10.3390/microorganisms9102019

**Published:** 2021-09-24

**Authors:** Xiao Song, Junting Song, Qi Yan, Jin Zhou, Zhonghua Cai

**Affiliations:** 1School of Life Sciences, Tsinghua University, Beijing 100084, China; song-x15@tsinghua.org.cn (X.S.); songjt20@sz.tsinghua.edu.cn (J.S.); yanq18@mails.tsinghua.edu.cn (Q.Y.); 2Shenzhen Public Platform of Screening and Application of Marine Microbial Resources, The Shenzhen International Graduate School, Tsinghua University, Shenzhen 518055, China; zhou.jin@sz.tsinghua.edu.cn

**Keywords:** oyster culturing, benthic microbial community, stochastic and deterministic processes, keystone species

## Abstract

The introduction of oysters to a waterbody is an efficient method for decreasing levels of eutrophication. Oysters affect sedimental environments and benthic microbes via their roles in nutrient cycling. However, little is known about how long-term oyster culturing affects benthic microbial community assembly. In the present study, top and bottom sediments from an oyster-culture area and non-culture area, in a eutrophic bay with a long history of oyster culturing, were obtained for environmental parameter measurement and microbe identification. Deterministic and stochastic processes in microbial community assembly were assessed. In particular, keystone species identification through network analysis was combined with measured environmental parameters to determine the factors related to community assembly processes. Our results suggest that oyster culturing relates to greater variation in both biological and non-biological sediment profiles. In benthic communities, *Proteobacteria* and *Chloroflexi* were the most abundant phyla, and community compositions were significantly different between sample groups. We also found that community assembly was more affected by deterministic factors than stochastic ones, when oysters were present. Moisture, or water content, and pH were identified as affecting deterministic and stochastic processes, respectively, but only water content was a driver associated with oyster culturing. Additionally, although keystone species presented a similar pattern of composition to peripheral species, they responded to their environments differently. Furthermore, model selection, fitting keystone species to community assembly processes, indicates their role in shaping microbial communities.

## 1. Introduction

Eutrophication is a long-term problem, threatening aquatic environments worldwide [1,2,3,4]. Various suspension and deposit-feeding bivalves, such as oysters [5,6], clams [7], and mussels [8,9], are among the species employed to address eutrophication. These creatures serve to filter particles from their aquatic habitats by suspension-feeding [10], which decreases the levels of nutrients in and increases the transparency of the water [11]. In addition, bivalve deposition can change sediment fluxes of nutrients by influencing nitrification and denitrification of the associated microbes [12,13]. The role of these shellfish in aquatic ecosystems varies with different environmental conditions [14], species [15], and the methods of cultivation. Hanging-cultured oysters act as connecters that link the water column and sediment. Thus, biodeposition by oysters may change benthic nutrient composition [5,16]. Dynamic sedimental environments, in turn, affect the associated microbes and thus their metabolism and function in biogeochemical cycling [17]. Marine microbes are the most important constituents driving element circulation and energy exchange [18]. Oysters have been reported to significantly change their environment and its related microbe community composition [19]. However, further studies are necessary on how oyster culturing affects microbial community assembly, particularly in accumulated profiles.

Microbial community assembly, generally, depends on both decisive processes, when the community is under selection by environmental condition, and neutral processes, when the composition of the community is random [20]. These processes are divided into four types—selection, drift, speciation, and dispersal—as drawn from Vellend’s conceptual framework [21]. Null model methods have been used to quantitatively disentangle stochastic processes from deterministic ones [22,23]. In eutrophic bays, microbial communities may be shaped by both deterministic and stochastic processes. The deterministic processes relate to strong selection pressure imposed by both heterogeneous nutrients and environmental gradients, from nearshore to farther out in a bay [24], which are usually markers of heterogeneous and homogenous selection, respectively. Furthermore, sediment microbes in shallow bays are affected by neutral processes associated with tides, waves, and current mixing [25]. In other conditions, however, sediment blocks the exchange of microbial communities between distant sites [26]. In these cases, homogenizing dispersal and drift affect community assembly processes.

Except for environmental conditions, microbial-community composition and assembly can also be affected by inter-species relations. Network analysis, based on Pearson correlation coefficients, has revealed the relationship between microbial community members. Molecular ecological networks (MENs) [27] offer a non-arbitrary way of setting a correlation coefficient threshold, and modularity has been introduced to describe the structure of a given network. Keystone species identified by network analysis fall into three types: the connecters between modules, the hubs or organizers of modules, and species with both roles. Such keystone species are the most important members in a community and their lack would lead to a breakdown of their community [28]. However, process analysis has not included inter-species relationships as factors that impact the turnover of a microbial community.

Shenzhen Bay (Guangdong Province, China), a shallow, narrow, semi-enclosed bay, is among the most eutrophic water bodies in the world because of rapid urbanization in surrounding areas. It has been a traditional oyster aquaculture area and more than 10 km^2^ of water is there employed for oyster-culturing rafts, with about 120 tons of production per year (https://www.afcd.gov.hk/english/fisheries/fish_aqu/fish_aqu_mpo/fish_aqu_mpo.html; accessed on 14 September 2021). It is believed that oyster culturing may be related to microbial-community composition in Shenzhen coastal waters [19].

Here, we measured the environmental parameters and microbial composition in sediments of the inner part (closer to the land) of Shenzhen Bay, from both culture and non-culture areas, to explore the effects of oyster culturing on the benthic microbial community. We hypothesized that (1) in the culture area, microbial assembly is more affected by deterministic processes, and (2) keystone species are an important factor in shaping the composition of the entire community. We aimed to gain a better understanding of the impacts of long-term oyster-culturing on the benthic microbial community in a typical eutrophic inner bay, which may provide a new perspective for assessing the ecological value of the oyster-microbe system.

## 2. Materials and Methods

### 2.1. Study Sites and Sample Collection

Shenzhen Bay is a typical urban bay located between two metropolises, Hong Kong and Shenzhen, and it has been suffering from eutrophication and other environmental problems resulting from heavy nutrient-loading and land-reclamation projects during urban development [29]. A considerable area of the south part of the bay (the Hong Kong side) has been employed for oyster (*Crassostrea rivularis*) farming by hanging clusters on rafts, and this practice has been ongoing for 200 years. By contrast, the Shenzhen side of the bay has not been suitable for aquaculture, since at least 2004, according to the government gazette of Shenzhen (http://www.sz.gov.cn/zfgb/2004/gb412/content/post_4968010.html; accessed on 20 February 2020). Thus, we identified two study areas: the oyster-culture area on the Hong Kong side and, on the Shenzhen side, a non-culture control (Figure 1). In January 2019, 16 independent sediment cores were collected in Shenzhen Bay with a home-made tube-shaped acrylic sediment core sampling device (Appendix A, 7-cm inner diameter and 120-cm long). Eight cores were taken from the oyster-culture area and another eight from the non-culture area. Adjacent samples from the same side were about 30 m apart. For each column, the top and bottom layers (0–30 cm and 61–90 cm depth below the water-sediment interface, respectively) were taken, avoiding the inside wall of the sampling tube, and kept in a cooler with vacuum bags after being subsampled for microbial determination (described below). Because each area (culture and control) included eight cores, there were four groups (the top layer of the oyster-culturing area, TO; the bottom layer of the oyster-culturing area, BO; the top layer of the non-culture area, TN; the bottom layer of the non-culture area, BN) of samples with eight replicates per group. The number of replicates here satisfied the sample size required for this kind of method [30] and has been adopted by previous research [31].

More than 10 g of each sediment was collected in pre-sterilized tubes, taken to the laboratory in liquid nitrogen and kept in a −80 °C refrigerator before DNA extraction. Sediments were then subsampled for measurements of water content, ammonium concentration, and proportions of organic carbon and nitrogen in the laboratory, within 6 h of sampling.

### 2.2. Environmental Parameters

Sediment parameters of pH and ORP (oxidation-reduction potential) were measured in situ using a multi-parameter meter (Hach HQ40d Dual-input). The parameters of moisture, NH_4_^+^, total nitrogen (TN%), and total organic carbon (TOC%) were measured in the laboratory. Briefly, the moisture of the sediment was determined by weighing before and after drying, following the state standard methods (GB17378.5-2007, China) with three subsamples from each sample. To determine the concentration of NH_4_^+^ in the sediment pore water, a triplicate of subsamples was freeze-dried before being mixed with ultrapure water (1 g dry weight sediment/5 mL water) and shaken vigorously. The slurry was centrifuged to obtain the supernatant, and all liquid samples were filtered through a 0.2-μm glass-fiber filter and tested on an automatic discontinuous chemical analyzer (Cleverchem 380, Dechem-Tech, Hamburg, Germany) using standard methods [32] within 24 h after sampling. The proportions of total nitrogen (TN%) and total organic carbon (TOC%) in the sediments were measured on a MACRO Cube Elemental Analyzer (Elementar, Lomazzo, Italy) [33] according to the manual. To be brief, samples were subjected to HCl (0.5 M) overnight, rinsed with ultrapure water three times, and dried in an oven at 60 °C until their weights remained stable for 1 h. Dried samples were ground and sieved through a 150-mesh sieve before measurement.

### 2.3. DNA Extraction, PCR Amplification, Pyrosequencing, and Data Processing

Total DNA was extracted from the sediments using a Fast DNA Spin Kit (mBio, New York, NY, USA) according to the manufacturer’s instructions. The extracted DNA was dissolved in 100 μL of Tris-EDTA (TE) buffer, quantified based on optical density (OD) using the NanoDrop One (Thermo Fisher Scientific, Waltham, MA, USA) spectrophotometer (OD_260_/OD_230_ > 1.8), and stored at −20 °C until further use. V4-V5 regions of prokaryote 16S rRNA genes were amplified using the primer pair 515F (5′-GTGCCAGCMGCCGCGGTAA-3′) and 907R (5′-CCGTCAATTCMTTTRAGTTT-3′), synthesized by Invitrogen (Invitrogen, Carlsbad, CA, USA). PCR reactions, containing 25-μL 2x Premix Taq (Takara Biotechnology, Dalian Co. Ltd., Dalian, China), 1 μL of each primer (10 mM), and 3 μL DNA (20 ng/μL) template in a volume of 50 μL, were amplified by thermocycling: 5 min at 94 °C for initialization, 30 cycles of 30 s denaturation at 94 °C, 30 s annealing at 52 °C, and a 30-s extension at 72 °C, followed by final elongation at 72 °C for 10 min. The PCR instrument was BioRadS1000 (Bio-Rad Laboratory, Billerica, CA, USA). A single composite sample was prepared for pyrosequencing by combining approximately equimolar amounts of PCR products from each sample. Sequencing was performed at Guangdong Magigene Biotechnology Co., Ltd. (Guangzhou, China) on an Illumina Hiseq2500 platform, and 250 bp paired-end reads were generated.

Paired-end raw reads were trimmed and filtered according to the Trimmomatic (V0.33, http://www.usadellab.org/cms/?page=trimmomatic; accessed on 13 March 2019) quality-controlled process (score: 20; cut off: 100) [34]. The generated paired-end clean reads were merged using FLASH (V1.2.11, https://ccb.jhu.edu/software/FLASH/; accessed on 13 March 2019) [35], with at least 10 bases overlapping and an error ratio of the overlap region of no more than 0.1. Sequences were assigned to each sample, based on their unique barcodes and primers, using Mothur software (V1.35.1, http://www.mothur.org; accessed on 13 March 2019) [36] before removing barcodes and primer sequences. Sequences were assigned to operational taxonomic units (OTUs) using USEARCH Software (V10, http://www.drive5.com/usearch/; accessed on 13 March 2019) [37] with a similarity threshold of 97% and chimera sequences and singleton OTUs removed. For a representative sequence of each OTU, the Silva (https://www.arb-silva.de/; accessed on 13 March 2019) [38] database was used to annotate taxonomic information. Unclassified and chloroplast- or mitochondria-related OTUs were removed before an abundance table was constructed and normalized. Additionally, QIIme (http://qiime.org/scripts; accessed on 13 March 2019) [39] was used to generate the phylogenetic tree. The sequence data reported here have been deposited in the NCBI GenBank database under the accession number SAMN16408193.

### 2.4. Data Analysis

All statistical and other analyses of environmental and biological data were performed with R (R version 3.4.0) unless another software package is specifically mentioned. The significance level was set at a *p*-value below 0.05 unless stated otherwise. The main analysis methods are summarized in Appendix A.

#### 2.4.1. Environmental Parameters

Each measured environmental factor was checked for normality within each group using the Shapiro–Wilk test. Levene’s test, in the R package “car”, was used to test for homogeneity of variance. If the variances were equal, a Turkey test was performed. When the variances were not equal, ANOSIM was applied on all environmental parameters followed by the Steel–Dwass post-hoc test. Subsequently, every parameter was scaled by subtracting the minimum value then dividing by the gained maximum for further analysis. Principal component analysis (PCA) was applied to the scaled data. Correlation analysis of environmental indexes (pre-processed) using the method “Pearson” was performed to test how these factors related to each other, across all samples or within areas.

#### 2.4.2. Community Biodiversity

The abundance table, based on OTUs, was first re-sampled to get a working draft table with each sample having the same number of individuals. An ANOSIM test was performed on the working table to determine if community composition was significantly different among groups. When they were, the Wilcox test was applied to test the significance of differences between each pair of groups. A Venn diagram was used to present group specific and shared species in terms of groups. We calculated the relative abundance of every OTU for each group and converted the data into presence–absence data. A species was “present” in a group when it was present in no less than half of the samples (that is: ≥4) of that group. Otherwise, we considered it absent from the given group. The α-diversity (indicated by the Shannon–Wiener index) and β-diversity (based on Bray–Curtis distances) were calculated. β-diversity was visualized through mon-metric multidimensional scaling (NMDS) analysis. Additionally, we calculated βMNTD (β-mean nearest taxon distance) as phylogenetic β-diversity and visualized it through principal coordinate analysis (PCoA). The α-diversity and phylogenetic and non-phylogenetic β-diversity were also calculated for keystone species identified from network analysis, described below. In addition, a rarefaction curve for each sample was calculated, based on the Chao1 index to show whether the sample sequencing volume was sufficient.

#### 2.4.3. Deterministic and Stochastic Processes

We calculated βNTI (β-nearest taxon index) and RC_bray_ to decide the relative contributions of deterministic and stochastic processes in the assembly of microbial communities in the sampling locations. βNTI describes how much the βMNTD deviates from a null model, assuming that all regional communities have no significant phylogenetic difference. RC_bray_ is used to infer whether the actual β-diversity was significant under the hypothesis of a moderate dispersal. βNTI values ≥2 or ≤−2 indicate significantly higher or lower turnover of a community compared with the expected value for a stochastic scenario; βNTI between 2 and −2 suggests the prevalence of stochastic processes. RC_bray_ values higher than 0.95 or lower than −0.95 are considered to represent significantly higher or lower turnover than expected under a moderate rate of dispersal; RC_bray_ between 0.95 and −0.95 indicates that the dispersal rate is moderate. We inferred these scenarios, according to standard approaches and that of Tripathi et al. [40], as homogeneous selection (βNTI ≥ 2), heterogeneous selection (βNTI ≤ −2), drift under dispersal limitation (−2 < βNTI < 2 and RC_bray_ ≥ 0.95), homogenizing dispersal (−2 < βNTI < 2 and RC_bray_ ≤ −0.95), and undominated (−2 < βNTI < 2 and −0.95 < RC_bray_ < 0.95), the latter meaning that none of these processes is significant. The relative contributions of processes were calculated based on these two indices at three levels: regional, local, and between–local (more details are provided in the Results section). βNTI and RC_bray_ were also tested for keystone species (see below).

It should be noted that Shenzhen Bay experiences a flood season and a dry season every year. Higher precipitation during the flood seasons obviously introduces heavier impacts on the processes of microbial community assembly. Although both sites in the present study are equally affected in the flood season, it is possible for the effects of oysters to be overwhelmed.

#### 2.4.4. Network Analysis

For each group, a network was constructed from the working abundance table to show the relationship between species within each environment, using MENs, through Random Matrix Theory (RMT)-based methods [27]. Each network was constructed into modules and its characteristic indices were calculated. Moreover, keystone species were identified with Z_i_ and P_i_ values. These were calculated through an online analysis platform (http://ieg2.ou.edu/MENA; accessed on 13 March 2019). Here, Z_i_ describes the importance of any node in a module, while P_i_ tests the rate of within- and inter-module links of a node. Nodes with Z_i_ values higher than 2.5 were identified as module hubs or module organizers; nodes with a P_i_ value over 0.62 were termed the connectors; nodes playing both roles were termed network hubs or network organizers. All three types of nodes were classified as keystone species and the other nodes as peripheral species.

#### 2.4.5. RDA and Model Selection

Redundancy analysis (RDA) was performed for three sets of OTUs (A: all species, B: keystone species, and C: all but keystone species) to compare how different sets of species react to environmental factors. As keystone species usually comprise a small proportion of a community, we tested the results by taking 99 random samples with the same number of individuals as keystone species from the community. Note that set C of OTUs included both peripheral species and species that were not included in the network.

The environmental parameters and keystone species that related to deterministic and stochastic processes, respectively, were identified according to Stegen [41]. Environmental parameters and keystone species were combined through PCA. The generated PCA axes were then fitted to βNTI or RC_bray_ through a distance-based redundancy analysis (dbRDA) and were filtered with the model selection process (ordiR2step). The retained PCA axes were identified as being related to deterministic or stochastic processes. PCA loadings for each environmental parameter or species abundance were calculated, and the parameter with the highest loading on a retained axis was identified as a primary factor in the process. That is, if a factor had the highest loading on a PCA axis, retained through model selection based on βNTI, it was considered related to selective processes, and factors with the highest loading on a PCA axis retained from RC_bray_ but not βNTI model selection were identified as being related to dispersal.

Pearson correlation coefficients were calculated for each taxa–environmental factor pair.

#### 2.4.6. Water Parameters and Sedimental Acid Volatile Sulfides (AVS)

On the sampling days, another study in Shenzhen Bay was performed in the same area—several water parameters (chlorophyll, conductivity, temperature, and dissolved oxygen were measured in situ with EXO I (YSI, Yellow Springs, OH, USA), and extra sediment samples were taken from the same areas as used in the present research for AVS detection (unpublished data). ANOSIM and PCA were performed on water parameters after scaling. Further, a heat map was created, based on AVS and other sedimental data, to investigate whether keystone species were significantly related to environmental factors.

#### 2.4.7. Genome Prediction

PICRUSt [42] was performed to predict functional genes for the microbial community of the TO group. Another genome prediction tool, Tax4Fun [43], was applied to keystone species.

## 3. Results

### 3.1. Environmental Parameters

All environmental indexes (Appendix A and Figure 2) varied notably across at least some groups. Most parameters exhibited significant differences between culture and non-culture areas (Figure 2A–F), with pH the only exception, showing a difference between depths but not areas. A similar inter-group pattern was observed for ORP, moisture, TN%, and TOC% (Figure 2B,C,E,F). For these parameters, the oyster-culturing site exhibited the highest value in the top layer and the lowest in the bottom, while no significant difference was observed between depths in the control area except for TOC%. Ammonium showed the opposite trends in the culture and the control areas, and TO showed the highest NH_4_^+^ (Figure 2D). ANOSIM on measured abiotic factors showed significant differences across the four groups (R = 0.665, *p* = 0.001) (Appendix A) and all post-hoc tests between groups were significant.

Cluster analysis based on environmental parameters showed that the TO and BO samples were clustered together, while samples from TN and BN group cannot be separated (Appendix A). In the PCA based on measured environmental factors (Figure 2G), the PC1 and PC2 axes summed accounted for 80.6% of the difference among samples. Samples from the culture site were separated, in terms of depth, along the first axis (62.7%), to which TN%, TOC%, moisture, and ORP were oriented, while control samples from different depths were loaded on the second axis (17.9%), to which ammonium and pH contributed. Sample points representing the culture area were more scattered compared with those from the control area, indicating more variable environments. As some factors seemed to be strongly related to each other, a more detailed analysis, focusing on correlations, was performed (Appendix A). Interestingly, although environmental parameters were strongly related to each other within samples from the culture area (Appendix A), their correlations were much weaker across the control samples (Appendix A). In the area without oysters, the only significant correlation was between TN% and TOC%.

### 3.2. Community Composition and Biodiversity

Rarefaction curves showed that the sampling volume was reasonable (Appendix A). The relative abundances of bacteria in the sediments suggested *Chloroflexi* and *Proteobacteria* as the most abundant phyla, followed by *Planctomycetes*, *Bacteroidetes*, and *Acidobacteria* (Appendix A). A total of 12,293 original OTUs were identified from clean data. They were then randomly sampled to generate the 1195 most abundant OTUs (more than 80% of the reads). The ANOSIM performed on the working table suggested that the microbes of the four groups had significantly different composition (R = 0.766, *p* = 0.001) (Appendix A).

The α-diversity of the microbial communities was higher in shallower sediment and when oysters were present. Thus, TO sediments supported the highest microbial diversity, while BN represented the lowest (Figure 3A). A Venn diagram also showed that 5 group-specific species were identified in the TO group, and no specific species in other group (Appendix A). Benthic community composition was notably explained by sediment depth, while the presence of oysters played a less important role (Figure 3B). However, when phylogenetic relationships were taken into account, microbial-community composition significantly responded to both depth and oyster culturing (Figure 3C). Additionally, communities from the culture area were more affected by depth than were those from the control site.

Rarefaction curves, based on Chao1, for each sample showed that the sample sequencing volume was sufficient.

### 3.3. Deterministic vs. Stochastic Processes

Stochastic processes played a more important role in regional community assembly but deterministic ones made nearly comparable contributions (Figure 4). Within 496 sample pairs, 216 pairs (43.5%) were affected mainly by heterogeneous selection, 240 (48.4%) by dispersal limitation, 5 (1%) by homogenizing dispersal, and 35 (7.1%) were not dominated by any single process. Homogeneous selection did not dominate any sample pair. However, contributions of these processes were different in the four groups. TO was more affected by heterogeneous selection (60.7%) while TN by dispersal limitation (64.3%). In the bottom layers, deterministic processes contributed more than stochastic ones when oysters were present (54.5 to 33.2%) but less so when they were not (17.9 to 63.5%). Process analysis between groups suggested different contributors to community turnover. Differentiation of communities between the culture groups TO and BO was mostly affected by heterogeneous selection (56.3%) and less affected by dispersal limitation (43.7%). As for the control area, heterogeneous selection (40.6%) played a less important role compared to dispersal limitation (53.1%). These results suggest that environmental selection became a greater driver for shaping the microbial community when oysters participated. Dispersal limitation impacted community structure between two groups at the top of the sediment (81.9%) but did not affect the bottom groups as much (40.6%). Heterogeneous selection, on the other hand, made a smaller contribution between the groups’ top layers (18.9%) than between their bottom ones (50%). The βNTI and RC_bray_ are listed in the supplementary materials (Appendix A).

### 3.4. Network and Keystone Species

Global indices of network showed that the microbial network from the TO group was most modular, which means that intermodule connections were relatively few compared with intramodule ones (Appendix A). There were 11 species identified as module organizers, and no connecters were identified in the TO network. For the BO, TN, and BN groups, 8, 14, and 9 species were recognized as module organizers and 7, 4, and 3 species as connectors, respectively. There were no network-organizing species in any group. Different sets of keystone species were identified from respective networks, except for a couple of overlaps. Thus, 52 keystone species in total were identified, most of which were in the phyla *Proteobacteria* (19 OTUs), *Chloroflexi* (13 OTUs), or *Bacteroidetes* (7 OTUs) (Appendix A).

The diversity of these 52 keystone species (Figure 5A) exhibited a similar pattern to that of all 1195 species (Figure 3B), while the phylogenetic composition of the keystone species (Figure 5B) showed differentiation by depth only, and not by area. The relative contributions of the processes affecting the keystone species (Figure 6) varied considerably compared with when those of the whole community. Generally, only a fifth of the variation in these keystone members was related to heterogeneous selection, and drift under dispersal limitation affected less than three in ten of the possible sample pairs, leaving more than half of the overall differences not dominated by any process. For most within- and inter-group comparisons, more than half of keystone species turnover was not dominated by any process. Two exceptions occurred within the BO group and between BO and TO groups, with a considerable contribution of drift (43%) was observed in the former and an even greater dispersal limitation (58%) in the latter. Unlike circumstances for all species, the exchange of keystone species between the two sites was less limited in the top (3%) than in the bottom layer (16%). The βNTI and RC_bray_ based on keystone species are listed in the Appendix A.

### 3.5. Keystone Species vs. Other Species

Community composition was explained by measured environmental factors for three sets of species: the whole community (A), keystone species (B), and the whole community without keystone species (C) (Appendix A). The total proportion explained by environmental factors (all six RDA axes) for keystone species was 61.1% (Appendix A), higher than that for the other two sets (ll: 58.3% (Appendix A); non-keystone: 57.7% (Appendix A)). However, when the whole community was sampled randomly to form 99 artificial sub-communities with the same number of individuals as keystone species (52 OTUs), the community composition as explained by environmental factors showed no significant difference between these subcommunities and keystone species. Keystone species from the BO responded more to measured factors than did keystone species of the other three groups (Appendix A). Sets of keystone species were more related to environmental conditions than were sets of all species in the BO, TN, and BN groups, to which the TO group was the only exception (Appendix A). Species that were significantly related to environmental parameters accounted for a larger proportion in the bottom layers than in the top layers at either site, which was true for both keystone species and the whole community.

### 3.6. Model Selection

There were 32 PCA axes generated from PCA, based on a combination of environmental factors and keystone species. From these 32 axes, PC2, PC24, and PC8 were identified as accounting for selection and PC9, PC5, and PC11 explained dispersal of the benthic community. The environmental factor that most contributed to the selective process was pH (highest loading on PC2), while moisture (highest loading on PC5) was most strongly related to stochastic processes. Keystone species contributed to shaping the community through both deterministic and stochastic processes. The related biotic and abiotic factors are listed in Table 1. Model selection that fitted environmental PCA axes to βNTI and RC_bray_ based on keystone species showed TN% to be a deterministic process-related parameter, and pH and NH_4_^+^ concentrations as stochastic ones (results not shown).

### 3.7. Water Parameters and Sedimental AVS

ANOSIM (Appendix A) and PCA (Appendix A) based on measured water parameters suggested no significant difference between the culturing and the control areas. In addtion, certain keystone species were significantly related to sedimental parameters (Appendix A). Among all measured parameters, pH was significantly associated with the most species. There were three keystone species significantly related to AVS. Other factors (TN%, TOC%, and moisture) were also related with keystone species, but with lower statistical significance.

### 3.8. Genome Predicion of TO Community and Keystone Species Based on KEGG Database

Genome prediction suggested metabolism related genes were most abundant in TO group and amino acid-related gene category was the most important in metabolism (Appendix A).

The metabolism of keystone species did not show significant intergroup difference.

## 4. Discussion

Benthic microbial communities in oyster-culturing areas strongly affect the fate of sedimental nutrients [16], and an analysis of the relative contributions of selection and neutral processes has been important in understanding microbial community assembly. In this study, we used ecological process analysis to distinguish deterministic processes (selection) from stochastic ones (neutral) in microbial community assembly and network analysis to highlight the role that keystone species play in shaping the microbial community of eutrophic coastal sediments associated with long-term artificially-cultured oysters and their accumulated impacts.

According to the present research, oyster culturing is related to local benthic TOC% and TN% in the inner part of Shenzhen Bay. These bivalves have been reported to assimilate aquatic nutrients and then translocate them to sediments by consuming phytoplankton and excretion [44]. Considering that sampling in the present study was performed at distant sites (although from the same bay), oysters here may serve to concentrate nutrients from a water body into biodeposits, causing local TOC% and TN% to increase. According to Appendix A and our unpublished data, basic water parameters in situ showed no significant differences between the culturing and control areas, suggesting that the deviation in sediments may be caused by oysters. We also expected that the bottom part of the sediment would exhibit at least some accumulated effects. The concentration of NH_4_^+^ was extremely high in the TO samples but decreased in deeper sediment, suggesting more active consumption of nitrogen compared with the non-culture area. Oysters also contribute to greater environmental variability [24]. They may serve to introduce additional sources of environmental variation to the system and drive conditions in the top sediment to diverge even further from those in the bottom layer. River input is usually considered to be an important resource of nutrients [45]. Shenzhen River is the most polluted river flowing into Shenzhen Bay. The oyster-culturing area and the control area are mainly effected by the Shenzhen River, and their hydrodynamic conditions are similar [46].

Diversity of the benthic microbial community in Shenzhen Bay significantly increases when oysters are present, as has been reported previously [17,47], and *Proteobacteria* has been identified as a dominant phylum [48]. In our study, *Chloroflexi* was also among the most abundant phyla, after *Proteobacteria*.

Both stochastic and deterministic processes played important roles in benthic microbial community assembly. In the first hypothesis, we predicted that in the oyster-culturing area, microbial community assembly would be more affected by deterministic processes than stochastic ones. Additionally, according to the results, microbial community assembly in the control area responded more to stochastic processes than to deterministic ones and selection was less important in the top layer due to more variable micro-environments (Figure 4). When exposed to a mixture of nutrient resources, community assembly tends to shift from the stochastic to the deterministic pattern because the microbial community with higher diversity has a broader ability to decompose organic compounds [49]. In other studies, however, a higher level of nutrients is suggested to reduce the importance of deterministic processes because more diverse resources are available in the system [50]. The results of the present research fitted the latter condition, showing that more varied environments supported a higher microbial diversity.

In the culture area, an opposite pattern of microbial community assembly was observed. Nutrients and other environmental factors explained a considerable proportion of the microbial-community composition, indicating that oyster farming was related to deterministic processes. However, the only measured nutrient parameter retained for heterogeneous selection was pH, which was not significantly related to the presence of oysters. The pH has been reported as an important factor associated with microbial community assembly. A higher pH is observed in the deterministic processes in alkaline environments [51] and neutral pH is more related to stochastic ones. One reason is that extreme pH could act as a strict selector to introduce homogenous selection (βNTI less than −2) [40]. However, sediment pH, here, was negatively related to heterogeneous selection. This means that, in the present study, sediments with neutral pH could support a microbial community with higher genetic and functional diversity. We took our samples from the inner part of Shenzhen Bay. In this area, pH was closer to neutral, which suggests less contributed seawater in the water body and a higher stability that could, in turn, diminish the stochastic aspects of microbial community assembly. Notably, pH in our study was close to 7.0, indicating that the role of pH in affecting community assembly may vary for different pH levels or ranges. Higher moisture levels in the soil are often associated with stronger dispersal because porewater can serve as a medium for exchange between separate communities [41,52]. Our results also showed that moisture had a negative relationship with dispersal limitation, and that sediments from the culture area contained more water.

Network analysis was employed to describe the global features of a given microbial community and its interspecific relationships. More network complexity means more interspecific connections and is usually observed within multiple environmental gradients [24], which fits the case of the non-culture site of the present study. However, the network was simplified when oysters were present, suggesting that the impacts of oysters may overwhelm the complex conditions in the study area. Additionally, lower modularity is supposed to be observed in the top sediments, due to the presence of a stronger hydrodynamic force. However, the network of the TO group was more modular than that of the BO or TN groups, suggesting that cultured oysters may help to stabilize the network.

“Keystone species” were defined based on their close bonds to the rest of the community. Keystone species identified in the present study were representative of the community in terms of composition but not function. The hypothesis that these keystone species reflect the structure of the whole community was supported by the study results (Figure 3B and Figure 5A). The proportions of keystone-species composition explained by environmental parameters was not statistically lower than that for the whole community. Nevertheless, the examined parameters made different contributions to these two sets of microbes. Moreover, the lack of similarity between keystone species and the community in phylogenetic β-diversity also indicated that these two sets of microbes were subject to different environmental factors (Figure 3C and Figure 5B). One example was NH_4_^+^. In RDA, based on all species, the NH_4_^+^ arrow was oriented close to the second RDA axis, along which the samples were separated by sites (Appendix A); however, it did not play a comparable role in keystone-species composition (Appendix A). According to the process analysis and model selection, keystone species were more affected by stochastic processes, while selective processes played a greater role for the whole community. Different parameters took part in both deterministic and stochastic processes for the two sets of species.

Keystone species can serve to stabilize the network and changes in them would profoundly affect the structure of the whole community [53]. These species from various habits have been suggested as indispensable to the ecological function of the system [54]. In the present study, although keystone species responded to environments in the same way that the rest of the community did, they showed higher resistance to the presence of oysters than did non-keystone species. Keystone species have been demonstrated, in many studies, to support community structure in two ways. Some of them can deal with a wide range of substrates, while others affect the degradation of specific substrates in a given environment.

*Chloroflexi* and *Proteobacteria* were among the most abundant keystone species in the present study. Previously, *Chloroflexi* have been reported as keystone species in aquatic systems and sediments [55]. *Proteobacteria* were dominant in the sediment of an oyster farming area due to their abilities to regenerate nutrients [48]. Among *Proteobacteria*, *Deltaproteobacteria* (*Desulfobacterales*, especially) and *Gammaproteobacteria* were reported to be in abundance in polluted coastal areas and play an important role in network functions, according to a study focusing on an urbanized coastal estuary [56]. In the present study, these bacteria dominated the community in habitats with high nitrate and sulfate which may be related to their ability to obtain energy from N and S metabolism [57,58]. Some keystone species identified in the current study are associated with specific functions. *Dehalococcoides*, for example, are able to degrade PAHs (polycyclic aromatic hydrocarbons) [59]. *Anaerolineaceae* may also participate in PAH biodegradation through anaerobic pathways [60]. PAHs are a known contaminant in Shenzhen Bay and the recent increasing pace of its accumulation can be linked to the rapid urbanization of this city. PAHs are reported to be more abundant in sediment and suspended particulate matter rather than in water [61]. Moreover, correlation analysis based on keystone species showed that some members, such as *Pirellulaceae*, were significantly related to specific substances such as AVS (acid volatile sulfides) (Appendix A). *Pirellulaceae* are one of the dominant phylum in a sulfur-rich mangrove environment but are not present in local sulfur metabolism [62]. Detailed information on Kyoto Encyclopedia of Genes and Genomes (KEGG) orthologs focusing on the local microbial community is required to determine whether it contributes to sulfur metabolism in the specific environments of Shenzhen Bay.

The dispersal limitation between TO and TN was more restrictive than it was between BO and BN, despite the top layer being adjacent to the water column and containing more water. This means that other factors may impede the dispersal process and are different between the top and the bottom sediments. Thus, more parameters should be measured and put through model selection before understanding why limited dispersal dominated community turnover in the top layer. Nevertheless, keystone species showed a stronger exchange in the top than in the bottom layer. This once again shows the resistance of keystone species to environmental selection, because exchange between communities fitted a stochastic way. The different assembly patterns between keystone species and the entire community that we uncovered has, to our knowledge, not been studied in detail before. Further understanding of patterns of microbial community assembly would benefit from similar and extended studies based on similar and different environments.

In the present work, we found that oyster culturing in Shenzhen Bay related to the benthic microbial community assembly through both stochastic and deterministic processes, which responded to different environmental parameters. This suggests the necessity of analyzing ecological processes to identify the role of oysters in shaping sediment microbial communities in this area. Additionally, keystone species also affected microbial community assembly, and the function of the community seemed more stable in the presence of oysters compared with non-keystone species. Thus, a closer look into their respective roles in the community may reveal more details of the ecological function of the resident microbes. Quantitative methods can be adopted to identify the relative contribution of stochasticity and determinism in specific situations. However, it is still challenging to disentangle environmental and biological factors related to stochastic and deterministic processes. In our study, we presented a method to describe the impact of keystone species on the microbial community in terms of both community composition and assembly. It should also be noted that impacts of drift were difficult to directly identify. These impacts can only be estimated by calculating the extent to which dispersal process reverses the differentiation introduced by drift.

## 5. Conclusions

The effects of oyster culturing on benthic microbial-community composition has been widely studied to evaluate the ecological function of these bivalves when they are employed to deal with water eutrophication. However, information about community assembly can be very important for regulating the role of local microbes. In this study, we found that oyster culturing was related to more varied microenvironments and a higher diversity of microbial communities in the sediment, which can be related to a wider range of substrate-processing potential. Being functionally stable to oysters, keystone species can serve to strengthen microbial networks in the sediments, which is necessary for the community to function as a whole. Moreover, a deeper layer of sediment was included in the study to show the accumulating impacts of oyster culturing on the benthic community. There were a few issues that require further clarification. Here, oyster culturing has been shown to be connected to both stochastic and deterministic processes of benthic community assembly, but none of the deterministic parameters were relevant to oyster culturing. New research investigating more environmental parameters is needed. We identified several keystone species through network analysis, and their effects on community assembly were calculated. KEGG data, based on metagenomic analysis, are still required to understand their roles in maintaining the ecological function of the associated microbial community. It should also be noted that our study was based on the inner part of Shenzhen Bay and many researchers have shown that environments in the inner and the outer parts of this bay, separated by Shenzhen Bay Bridge, are significantly different. The present study sampled only the inner part of Shenzhen Bay; a comprehensive understanding of microbial community assembly in Shenzhen Bay can be achieved by applying these research methods to the bay’s outer reaches, which will enable comparisons to be made with the inner-bay samples. Nevertheless, the methods presented here offer a way to analyze the impacts of keystone species on microbial community assembly as well as their relative contributions compared with abiotic factors.

## Figures and Tables

**Figure 1 microorganisms-09-02019-f001:**
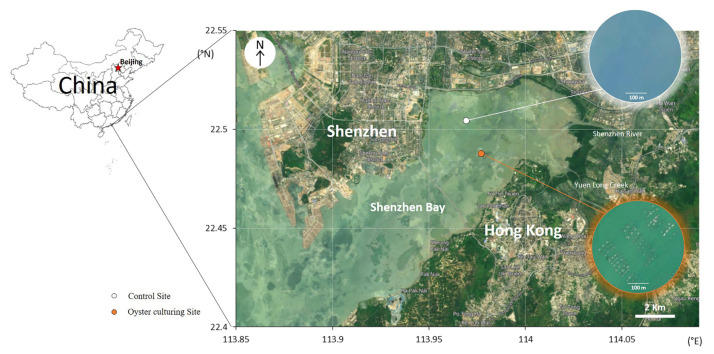
Location of the sampling sites. More detailed satellite images are inset on the right-hand side to show that there are many oyster-culturing rafts at the oyster-culturing site and no oysters being cultured at the control–non-oyster-culture site.

**Figure 2 microorganisms-09-02019-f002:**
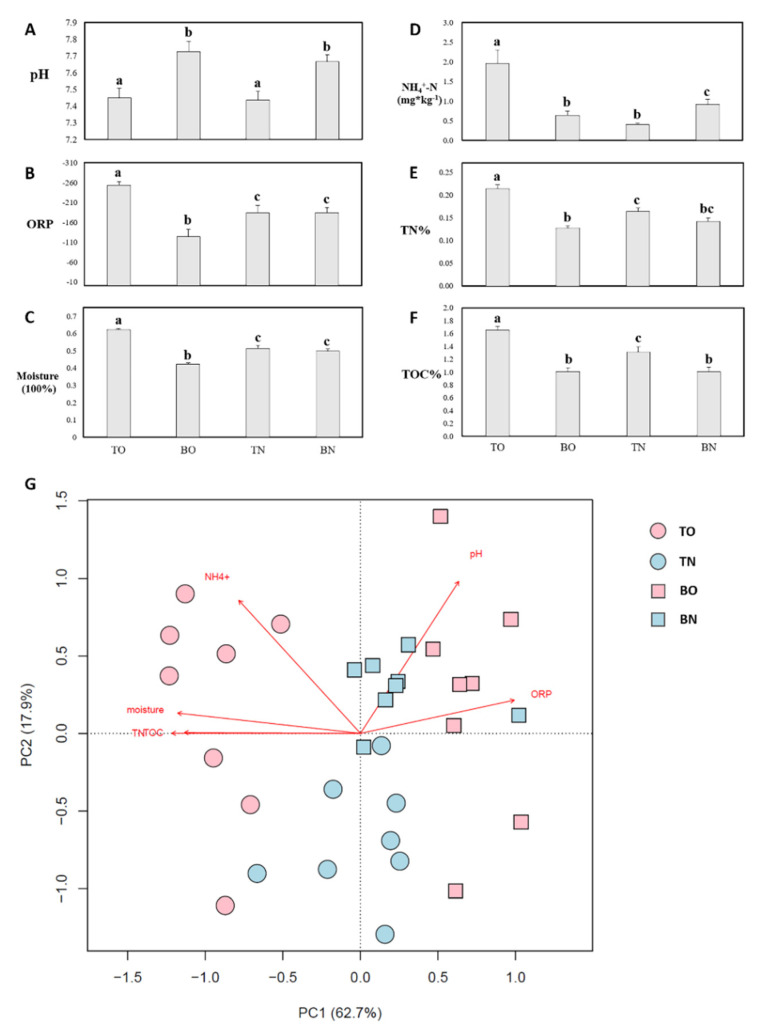
Environmental parameters. (**A**–**F**) Environmental parameters of the four groups. Groups sharing the same letter were not significantly different according to that parameter (*n* = 8, *p* < 0.05. Error bars, mean ± standard deviation); (**G**) PCA. Each point is one sample. Environmental parameters are shown as red arrows. Percentages in parentheses are proportional differences between parameters represented by the PC axis.

**Figure 3 microorganisms-09-02019-f003:**
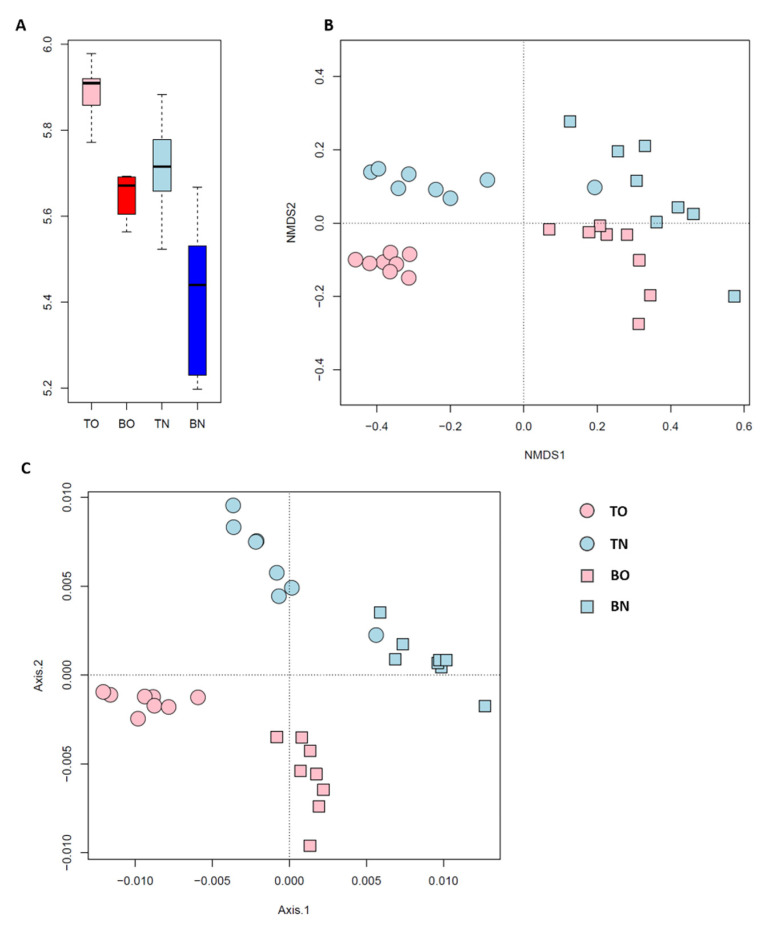
Diversity of benthic communities. (**A**) The α-diversity; (**B**) the non-phylogenetic β-diversity, visualized through NMDS based on Bray–Curtis distance; (**C**) the phylogenetic β-diversity, shown by PCoA. Each point in B and C represents one sample.

**Figure 4 microorganisms-09-02019-f004:**
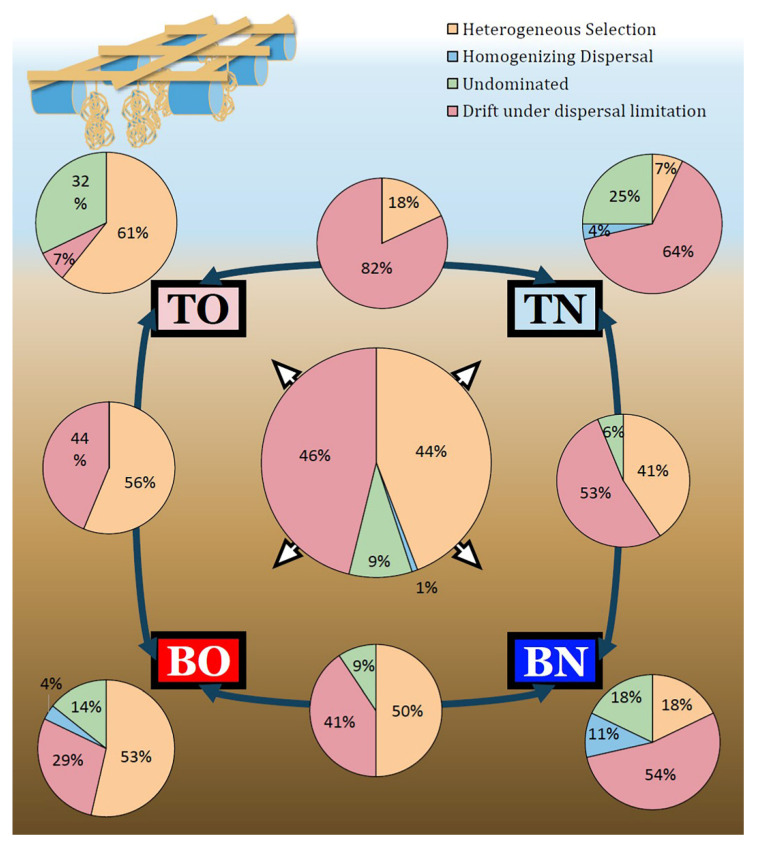
Contributions of different processes to community assembly. Sample groups are indicated by squares. Pie charts located at the four corners of the figure depict the relative contribution of processes in community turnover within groups. The pie charts on dark-blue double-headed arrows show the contribution proportions of various processes to the difference between the two groups connected by the arrow. The pie chart in the center indicates the contribution of processes in overall community assembly. (TO: the top layer of the oyster-culturing area; BO: the bottom layer of the oyster-culturing area; TN: the top layer of the non-culture area; BN: the bottom layer of the non-culture area).

**Figure 5 microorganisms-09-02019-f005:**
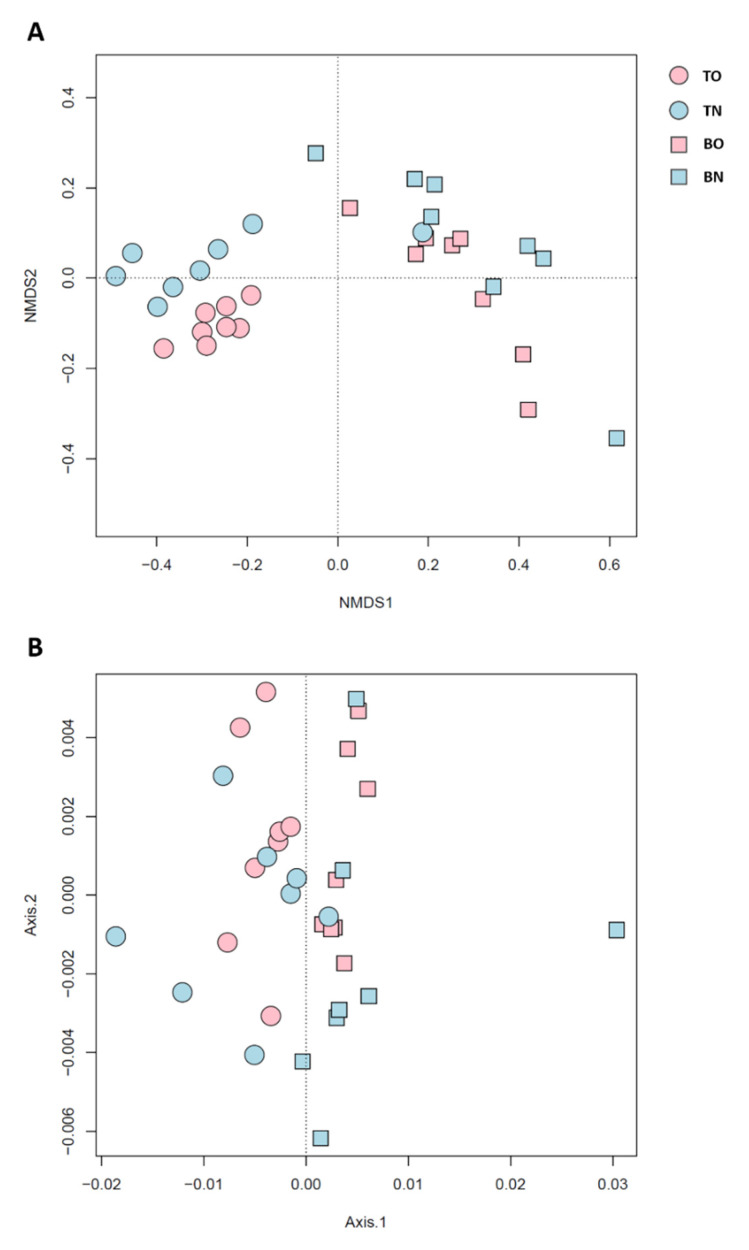
(**A**) Phylogenetic β-diversity based on keystone species only shown by NMDS; (**B**) keystone species based on phylogenetic β-diversity in the PCoA. Each point represents one sample.

**Figure 6 microorganisms-09-02019-f006:**
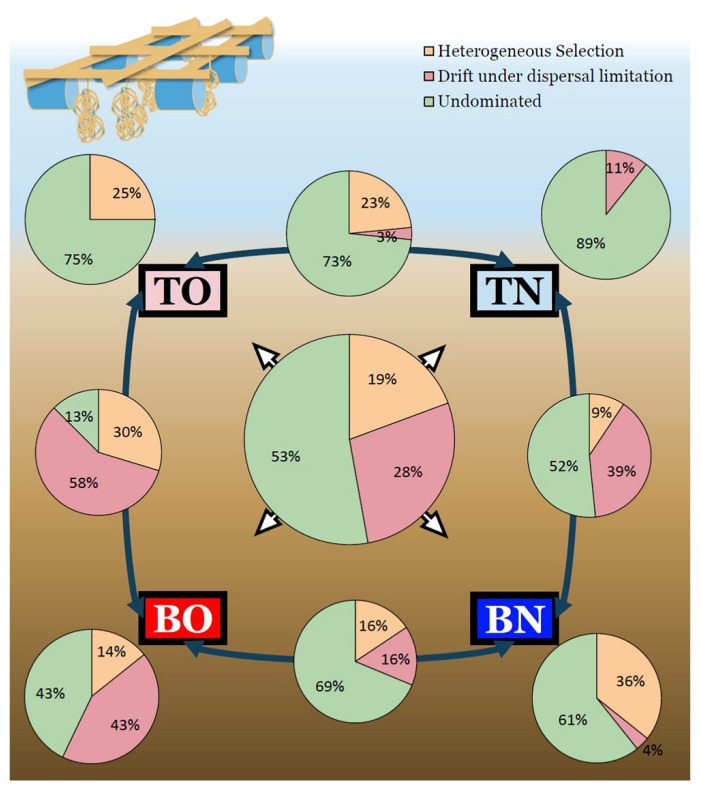
The contributions of different processes in keystone -pecies composition. Sample groups are indicated by squares. The pie charts located at each of the four corners depicts the relative contribution of processes in assemblies of keystone species within groups. The pie charts overlaying the midpoints of the dark-blue double-headed arrows show the contribution of various processes to the differences between the two groups (squares) connected by the aforementioned arrows. The center pie chart indicates the contribution of each process in the overall turnover of keystone species in the study. (TO: the top layer of the oyster-culturing area; BO: the bottom layer of the oyster-culturing area; TN: the top layer of the non-culture area; BN: the bottom layer of the non-culture area).

**Table 1 microorganisms-09-02019-t001:** The retained PCA axes through βNTI and RC_bray_ model selection and loadings.

	βNTI			RC_bray_		
	PC2	PC24	PC8	PC9	PC5	PC11
Environmental parameters						
Moisture	−0.1060	−0.0086	0.0457	−0.0512	−0.7764	−0.0177
NH4+	−0.6505	0.0077	−0.0378	−0.0210	0.3524	0.0147
ORP	**−0.1505**	−0.0074	−0.0674	−0.0279	**−0.3272**	−0.0171
pH	−0.7301	0.0036	−0.0538	0.0329	−0.1329	0.0037
TN	−0.0044	0.0097	**−0.2756**	−0.0019	−0.0968	−0.0316
TOC	−0.0033	−0.0055	0.3042	−0.0048	0.3193	0.0386
Keystone species						
OTU1	−0.0339	−0.1206	0.5251	0.3698	−0.1035	−0.0057
*Syntrophobacteraceae*						
OTU1764	0.0034	0.2667	−0.0029	−0.0150	−0.0196	0.0396
*Pirellulaceae*						
OTU18	0.0327	0.1811	−0.1745	0.1694	−0.0149	−0.2999
B2M28 *						
OTU1866	−0.0083	**0.2450**	0.0360	0.0331	−0.0024	−0.0653
ADurb.Bin180 *						
OTU209	0.0139	−0.0688	−0.1956	**0.2805**	0.0142	0.1885
Bacteroidetes_BD2-2						
OTU4	0.0177	−0.0827	−0.0212	−0.5950	−0.0233	−0.0096
*Anaerolineaceae*						
OTU44	0.0207	−0.0469	−0.2507	0.0377	0.0344	−0.4426
*Sulfurovaceae*						
OTU47	0.0132	−0.3834	−0.1244	0.1284	0.0185	−0.2602
*Chromatiaceae*						
OTU890	−0.0058	−0.0061	−0.0207	0.0891	0.0257	**−0.2895**
SBR1031 *						

The highest, second highest, and third highest loading on each PC are highlighted by black boxes, grey boxes, and bold text, respectively. * The level of class.

## Data Availability

Sequence data reported in this work have been deposited into the NCBI database under the accession number SAMN16408193.

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
