# Peer review of "Assembly of a Benthic Microbial Community in a Eutrophic Bay with a Long History of Oyster Culturing"

_microorganisms, 2021, doi:10.3390/microorganisms9102019_

Round 1

Reviewer 1 Report

I reviewed the revision of the manuscript, and all my comments have been delt with accordingly.

I also confirm that English language has been extensively revised and is now suited for publication.

I only detected one typo mistake: line 155-156: the "c" of trimmomatic has been cut of from the word and placed after the link of the software.

Author Response

Dear Reviewer 1,

Thank you for your time and all the valuable suggestions concerning our manuscript 1386570. These good advices have helped us improve our article a lot. Your new advice and our response are listed as below:

Point 1: Line 155-156: the "c" of trimmomatic has been cut of from the word and placed after the link of the software.

Response 1: We thank you for your carefulness and the problem has been fixed.

Reviewer 2 Report

Lines 74-75. You wrote production of 160 tons per year. Are you sure? The culture area is enormous (10 sq, km) it can't be to produce only 160 tons of oysters. If so, any influence of the oysters on bottom microbiota is negligible. Please check again.

Line 572. "is related"

In Figures 4 & 6 you could add a brief explanation of TO, TN, BO, BN in the caption in order to facilitate fast understanding by the reader. I think it will be better.

In Figure 4 it is better to indicate the open circle as "Control-non oyster culture site"

The rest are fine.

Author Response

Response to Reviewer 2 Comments

Dear Reviewer 2,

Thank you for your good comments and valuable suggestions concerning our manuscript 1386570. In the revised manuscript, we accept your advices and modified the text accordingly. We believe this helped us further improve our article. The detailed responses as below:

Point 1: Lines 74-75. You wrote production of 160 tons per year. Are you sure? The culture area is enormous (10 sq, km) it can't be to produce only 160 tons of oysters. If so, any influence of the oysters on bottom microbiota is negligible. Please check again.

Response 1: For this question, we checked both the area and the production. The area can be estimated using satellite images showing oyster drafts. However, these drafts are not close to each other (as showed in Figure 1), so the 10km2 is not the net area for oyster culturing but the size of the region which includes all the drafts.

As for the production of oyster, we check the web site of the Agriculture, Fisheries and Conservation Department of Hong Kong (https://www.afcd.gov.hk/english/fisheries/ fish_aqu/fish_aqu_mpo/fish_aqu_mpo.html).  According to the report, oysters require 4 – 5 years to grow before harvesting, and production of oyster is calculated as the weight of meat only (about 119 tons in 2020). We have included this web site address in the revised version.

Point 2: Line 572. "is related"

Response 2: The change has be made.

Point 3: In Figures 4 & 6 you could add a brief explanation of TO, TN, BO, BN in the caption in order to facilitate fast understanding by the reader. I think it will be better.

Response 3: Following sentence has been added to the captions of these two figures:

“TO: the top layer of the oyster culturing area; BO: the bottom layer of the oyster culturing area; TN: the top layer of the non-culture area; BN: the bottom layer of the non-culture area.”

Point 4: In Figure 4 it is better to indicate the open circle as "Control-non oyster culture site"

Response 4: Because only Figure 1 includes an open circle, we suppose you mean Figure 1? The change has been made accordingly.

This manuscript is a resubmission of an earlier submission. The following is a list of the peer review reports and author responses from that submission.

Round 1

Reviewer 1 Report

I had the pleasure of reviewing the manuscript entitled “Assembly of a Benthic Microbial Community in a Eutrophic 2 Bay with a Long History of Oyster Culturing” by Song et al.

The article is investigating the effect of oyster culture on the bacterial composition of the sediment at two different depths. They found that the oysters impacted the communities, by increasing its diversity its environmental variation.

I found the science sound; the experimental design and the statistical analyses are well explained, and the results and discussion are congruent with the findings. There is however an extensive work to be done on the English language, following the misuse of certain words, such as “endemic” (see below). There is also additional information/clarification that would be required in the material and methods.

I thus recommend accepting the paper, but with a major revision, especially for English language.

Introduction:

Line 50: an English sentence cannot start with “But”.

Line 68: replace “environments” with “environmental conditions”

Material and Methods:

Line 148-149: said “synthetized by Invitrogen” twice

Line 159-160: which filtering parameters were applied through trimmomatic (quality cut offs etc…)

Line 169: what is the version of the Silva database used?

Line 186: correlation analysis using which method? Pearson?

Results:

Line 297: The concept of “individual” here is not correct. I am guessing you want either to say “more than 80% of the original OTUs”, or “more than 80% of the reads of the initial dataset”.

Line 306: replace presented by “were present”.

Line 307: The term “endemic” here is misused. Endemicity is a different concept. Here I would say that some OTUs were “group specific” or “site specific”.

Discussion:

The discussion covers the key questions and findings of the study. However, it needs extensive English proofing.

Figures showing PCA: I would transform the figures showing PCA as they are not easy to understand at first. For better readability, use colors for the different sites (cultured versus non cultured) and shape for top or bottom layer. For example, red for cultivated area, blue for non cultivated area, and then circles for top layers and squares for bottom layers, for both sites. This will improve the visual.

Figure S5: caption to fix: replace “endemic” by “group specific”; replace “common species” by “shared species”.

Reviewer 2 Report

The authors compared the microbial diversity between oyster culturing area and non-culturing area based on various multivariate analyses. The authors conducted lots of analytical processes to support their hypothesis and it would be helpful for other researchers to analyze the microbial diversity within various habitat conditions using the similar analytical procedures. In order for it to be more informative, however, I have some comments below
I think the authors spends lots of efforts to describe the eutrophication in introduction and discussion. The results mainly focused on the description of the difference in microbial diversity between oyster culturing and non-oyster culturing regions, but it was difficult to find the results on the relationship with eutrophication and microbial community. So it is necessary to reduce the content of Eutrophication or add related results and discussion.
This manuscript applied lots of multivariate analyses so in the material and methods section, it would be good to add the analytical procedures as a figure to interpret easily the procedures. 
Sampling was conducted once in winter, and so it is necessary to add the sentences describing the sampling season selection and sampling frequency are representative to describe the microbial diversity in this region in the Methods section.
In lines 197-198, the authors describe the criteria for “present. Please add a relevant reference to the criteria.
The authors concentrated on the description of the difference between sampling points (TO, TN, BO and BN) through various analyses such as NMDS, RDA, etc but the description and discussion mainly focused on phylum or class not genus. For example, 2hy most of OTU about keystone species are phylum or class not genus or family? In NMDS and RDA, along with the ordinations for sampling points, it is necessary to add a description of the relationship between environmental factors and species with representing the species in the ordination map. Of course, it may be difficult to visually express all the microbial species that have appeared, so it would be good to add ordination for keystone species or species with high frequency when analyzing NMDs and RDAs. In addition, a keystone species was selected through network analysis, so it seems necessary to compare the functional and ecological characteristics of the keystone species among groups.
The authors use the word “endemic species” but what does endemic species mean here? Generally endemic species indicates the only species occurring in a certain country. If not, please choose another word to explain the species or describe the meaning of endemic species here. In addition, in the results of Venn diagram, the number of endemic species was higher in TO sample. Is there any special reason for it?
Please add the genus or family name instead of OUT information in Table 1 for the readers to interpret easily
In TO which have higher concentration of NH4+, is there any microorganisms related with the metabolism of NH4+?
The authors used many analyzes to show that there is a difference of microbial communities and environmental factors among TO, TN, BO, and BN. However, I think it is necessary to add cluster analysis based on microbial diversity data or environmental factor data before NMDS, RDA, etc to show whether there are significantly different among TO, TN, BO and BN.

Minor comments
Word unification is needed for P value. Some sentences have a “P” and some sentences have a “p”.
In the case of Figure 2 (D, E, and F), please unify the decimal places on the Y-axis.
It is described in the method, but for convenience, please describe what TO, TN, BO, and BN mean in the legend in Figure 2.
Line 404: specie -> species
In Table S2, it is necessary to mention the meaning of P value For example, *: >0.05, **: ? and ***:?
Table S5: What do the numbers in parentheses mean?
Need to add network analysis figure in supplementary.